# Engaging in Awkward Social Interactions in a Virtual Environment Designed for Exposure-Based Psychotherapy for People with Generalized Social Anxiety Disorder: An International Multisite Study

**DOI:** 10.3390/jcm12134525

**Published:** 2023-07-06

**Authors:** Pamela Quintana, Stéphane Bouchard, Cristina Botella, Geneviève Robillard, Berenice Serrano, Alejandro Rodriguez-Ortega, Mathias Torp Ernst, Beatriz Rey, Maxine Berthiaume, Giulia Corno

**Affiliations:** 1Département de Psychoéducation et de Psychologie, Université du Québec en Outaouais, Gatineau, QC J8X 3X7, Canada; 2Department of Psychology, University of Ottawa, Ottawa, ON K1S 5L5, Canada; 3Universitat Jaume, 12006 Castellón de la Plana, Spain; 4CIBER Fisiopatología Obesidad y Nutrición (CIBERObn), Instituto Salud Carlos III, 28031 Madrid, Spain; 5Departamento de Ingeniería Gráfica, Universitat Politècnica de València, 46022 Valencia, Spain; 6Department of Clinical Research, University of Southern Denmark, 5230 Odense, Denmark

**Keywords:** virtual reality, social anxiety, generalized social anxiety, exposure, cultural differences

## Abstract

The effectiveness of in virtuo exposure-based treatment of performance-only social anxiety disorder (SAD) has been demonstrated in several studies. However, few studies have validated virtual environments with participants suffering from generalized SAD. The goal of this study is to confirm the potential of a virtual environment in inducing anxiety in adults suffering from generalized SAD, compared to adults without SAD, when engaged in awkward social interactions. Differences between participants from two different countries were also explored. The sample consisted of 15 participants with SAD from Canada, 17 participants without SAD from Canada, 16 participants with SAD from Spain, and 21 participants without SAD from Spain. All participants were immersed in a control virtual environment and in an experimental virtual environment considered potentially anxiety-inducing for individuals with generalized SAD. As hypothesized, results showed that the experimental virtual environment induced a higher level of anxiety than the control environment among participants with SAD compared to those without SAD. The impact on anxiety of each socially threatening task performed during the experimental immersion was statistically significant. In terms of anxiety responses, no significant differences were found between participants from Canada and Spain. However, spatial presence and ecological validity were higher in Canadians than in Spaniards. Unwanted negative side effects induced by immersions in virtual reality were higher in the SAD group. This study highlights the importance for therapists to engage people with SAD in clinically relevant tasks while immersed in VR psychotherapeutic applications.

## 1. Introduction

Social anxiety disorder (SAD) is among the most common mental disorders in Western society after depression, alcohol dependence, and specific phobias [1]. Indeed, between 1% and 13% of the Western population meets the criteria for SAD at some point in their lives, with rather similar rates of 1.7% in Canada and 1.2% in Spain [2,3,4,5]. The intense, or marked, fear or anxiety of one or more social situations in which the individual can be observed, judged, embarrassed, or humiliated by others can have many consequences, such as social and occupational impairment. SAD is associated with elevated school dropout rates, lack of autonomy, and decreased well-being, employment, workplace productivity, socioeconomic status, and quality of life [4,6,7,8]. SAD is also associated with a higher probability of being single, unmarried, or divorced, and with not having children, particularly among men [6,7]. Furthermore, SAD often leads to the development of other psychiatric complications such as alcohol dependence [9], depression, suicide attempts, and other anxiety disorders [4,10]. Despite its severity and high prevalence, SAD remains largely undertreated [1,11,12]. Indeed, patients with SAD tend to avoid healthcare services as they do other social interactions [11,13]. They may feel ashamed of their symptoms and hesitant about discussing them even with healthcare professionals. Not seeking treatment often leads people with SAD to wait many years before consulting a health professional, by which time complications have often occurred [14].

Cognitive-behavioral therapy (CBT), either in an individual or group format, is the treatment of choice for SAD [15]. This therapeutic approach uses many techniques, but exposure is considered the main component responsible for treatment success [16,17]. This technique involves exposing, often gradually, the individual to anxiety-provoking stimuli in order to learn that the object of their anxiety or fear is not threatening [17,18]. In vivo exposure allows direct contact with the feared stimulus in real time; however, it raises challenges regarding the therapist’s control over the situation and the absence of privacy. The limitations of in vivo exposure can be overcome with an alternative medium of exposure: virtual reality (VR) technology or in virtuo exposure [19]. Fuchs et al. [20] define VR as the use of computer technologies and behavioral interfaces to simulate the behavior of 3D entities interacting in real time with each other and the user in pseudo-natural immersion via sensorimotor channels [20].

The most common way of experiencing VR is via a head-mounted display with built-in motion trackers and headphones. This equipment allows patients to practice exposure while limiting sound and peripheral vision from the surrounding environment [20,21], and VR allows therapists to conduct exposure exercises that ensure better confidentiality and control over stimuli [22]. Indeed, patient and therapist can work together in a controlled scenario where they can decide the intensity and duration of each exposure exercise. In some cases, in virtuo exposure-based treatment is financially less costly and requires less planning and preparation by the therapist than in vivo [23,24] because the therapist can control the behaviors of virtual characters according to the need for exposure (e.g., being attentive or distracted when needed) [25,26]. In addition, the use of VR can be beneficial to people with SAD as they are more reluctant to socially engage with people face-to-face than with virtual characters [27,28]. Patients can practice multiple exposure exercises during a single therapy session in the safety and confidentiality of the therapy room, reducing costs and increasing time efficiency [29]. In the last 20 years, an increasing number of empirical studies have proved the efficacy of using in virtuo exposure in the treatment of anxiety disorders [29]. Indeed, it has been demonstrated that in virtuo exposure-based CBT treatment is effective in treating anxiety disorders and represents several advantages over traditional CBT, making VR a powerful tool with which treatments are less aversive and more accessible. Furthermore, most studies have found no significant differences between the effectiveness of in virtuo exposure and traditional in vivo exposure [24,29]. These studies used real-time 3D rendering technology (i.e., computer-generated objects and virtual characters that are processed in real time by the computer as the user is interacting with them), sometimes including video loops of actors. Although VR does not create a perfect replica of the physical reality, people immersed in VR have the impression of really being in the mediated environment (sometimes referred to as spatial presence) and perceive this environment as lifelike and real (sometimes referred to as ecological validity) [18,19,20,21,22,24]. In their review on VR for the treatment of anxiety disorders, Anderson and Mollow [30] highlighted the critical importance of studying cultural issues and characteristics of virtual environments (VEs), as they have received almost no attention. Studying cultural differences when validating VEs for psychotherapeutic purposes recognizes the need to conduct research that is mindful of inclusive experiences. It provides empirical evidence of whether VEs are engaging for users, despite being developed in different cultural contexts, which has never been directly tested so far. Only with actual data will it be possible to know if VEs resonate with diverse audiences.

DSM-5 [6] distinguishes two types of SAD: the generalized type, characterized by the apprehension of almost all social situations in daily life, and the performance-only type, which is limited to excessive fear and anxiety in a performance situation (e.g., speaking in front of an audience). Generalized SAD differs from the performance-only subtype in that it has a higher number of feared social situations and involves more interactive social situations than talking to an audience. The effectiveness of using in virtuo exposure in the treatment of performance-only SAD has been demonstrated in several studies [24,27,31,32,33]. Clinical trials have also shown that in virtuo exposure using real-time 3D rendering technology is just as effective as traditional in vivo exposure when treating patients with SAD that have broader social fears than only performance [14,34,35,36]. However, VEs used for generalized SAD are often limited to performance situations, such as talking to an audience [37,38].

Research regarding in virtuo exposure-based CBT for generalized SAD remains limited, at least partly because VEs must recreate complex social contexts that often require dyadic interactions that were hardly possible decades ago using software designed for performance-only SAD. People with SAD are not afraid of other people but of the judgment of other people. Hence, immersion in VEs designed for exposure for SAD must not rely only on including virtual humans but on creating interactions where users could feel as if their behavior could trigger social judgment. Examples of situations that could trigger negative social evaluation include social interactions involving personal and intimate matters [39], non-verbal behavior expressed by virtual humans [40], being assertive, making blunders, or being involved in awkward situations [41]. The challenge in using VEs with people with SAD is therefore to engage in interactions in ways they could feel negatively evaluated by computer-generated characters.

The primary goal of this study was to test if engaging in awkward social interactions in a VE could induce anxiety in individuals with generalized SAD. The hypothesis was that awkward social situations occurring in a VE designed for exposure-based treatment of SAD would induce a higher level of anxiety, compared to a neutral VE, in people suffering from generalized SAD than in those from the control group. Furthermore, this study also explored potential cultural differences in anxiety intensity induced during an immersive experience designed for exposure-based treatment of SAD between a Canadian and Spanish population, as differences across countries are barely documented in clinical VR applications.

## 2. Materials and Methods

### 2.1. Participants

In total, 69 adults were recruited, comprising 15 participants from Canada with generalized SAD, 17 participants from Canada with no SAD, 16 participants from Spain with generalized SAD, and 21 participants from Spain with no SAD. To participate in the study, participants had to be (a) Canadian or Spanish, (b) 18 years of age or older, (c) born in their respective countries, and (d) raised by parents who were both born in the respective countries (Canada or Spain). They also had to identify themselves as native to the country (i.e., Canada or Spain) and be proficient in French or Spanish (oral and written). Individuals with a SAD score ≥4 according to the Anxiety Disorders Interview Schedule for DSM-IV (ADIS-IV; [42]) were considered clinical, and those with a SAD score <2 were considered non-clinical. The exclusion criteria were as follows: (a) have lived for more than one year in a country other than Canada or Spain; (b) have a relationship or had a spouse from a different country; (c) be an Aboriginal person from the respective country; (d) have a predisposition to unwanted negative side effects of immersions in VR (“cybersickness”, e.g., inner ear problems, repeated migraines, epilepsy, balance problems, eye problems, or severe and frequent travel/motion sickness problems); (e) meet the diagnostic criteria for performance-only SAD; (f) self-report having bipolar disorder; (g) have a significant self-reported neurological disorder; (h) self-report having psychotic symptoms; (i) take tricyclics (antidepressants), beta-blockers, or psychotropic substances on the day of the immersion (e.g., cocaine and ephedrine) that increase heart rate during the experiment; (j) self-report having stuttering difficulties [9]; and (k) have cardiovascular problems (e.g., arrhythmia). During recruitment, three Canadian and five Spaniard participants were excluded due to fulfilling criteria of performance-only SAD (response rate of 89% from the 77 adults screened). They were referred to other psychological support services.

The present study obtained the approval of the ethics committees of the institutions where the research was conducted (i.e., Canada: Université du Québec en Outaouais; Spain: Universitat Jaume I). Participation was voluntary and all participants provided written informed consent.

### 2.2. Procedure

Following ethics approvals received by both institutions, recruitment was carried out using posters displayed in the corridors and on the websites of the two universities and by advertising the research study in classes. First, a brief telephone interview was conducted to verify if potential participants appeared to meet the inclusion criteria (see Figure 1). For those who did not meet the inclusion criteria, a list of resources from local psychologists and a self-help manual were provided. Each selected participant was then invited to an individual meeting lasting two and a half hours. Upon arrival, participants were asked to read and sign the consent form and fill out a short demographic questionnaire (assessing age, sex, number of children, religion, previous experience with VR, and psychotherapy), followed by a clinical interview. Then, participants completed questionnaires and were introduced to VR through immersion in a control VE depicting an empty apartment. Their task was to explore the apartment for two minutes.

While preparing to be immersed in the experimental VE, the participant was informed that the immersion involved entering a convenience store, talking to the clerk, and asking for a refund for eggs. That convenience store had been the victim of numerous burglaries in the past year, so the clerk is now suspicious of customers and tends to refuse customers’ requests. The participant was told to initiate all dialogues, as the clerk was a bit grumpy and would wait for customers to engage conversation. The purpose of this instruction was to provide a narrative that would help create a context to facilitate presence and make some of the tasks more anxiogenic. No time was allocated for participants to prepare for the tasks performed in the VE. When immersed in the experimental VE, all participants were instructed to perform four tasks associated with social anxiety [41]. The first task’s goal was to generate anxiety about informal interactions. The participant was prompted to engage in a conversation by greeting the clerk in the virtual store and answering the clerk’s questions about the participant’s personal life (e.g., profession and intimate life with a partner). The second task was related to shyness and performance anxiety, where the clerk asked the participant to name as many Michael Jackson’s songs as possible. The third task was related to assertiveness anxiety and involved attempting to convince the clerk to exchange several eggs purchased from the store that had passed their expiry date. The clerk, expressing reluctance and frustration verbally and non-verbally, asked the participant to find better arguments to convince him. Finally, the fourth task was related to anxiety about being observed. When the virtual clerk agreed to exchange the eggs, he asked the participant to go pick up fresh eggs in the convenience store while he was observing the participant searching in vain as there were no eggs left in the store. All dialogues from the virtual human were pre-recorded (in French and Spanish), and the social interactions flowed naturally without pauses between the tasks.

As a token of appreciation, participants received an unpublished self-help manual for SAD, were briefed on how to use it, and received a list of references for local psychological resources.

### 2.3. Measures

References and descriptions of the measures refer to the French-Canadian and Spanish versions actually used. Physiological data (heart rate and skin conductance), as well as exploratory measures, were collected but not used due to methodological limitations [43].

#### 2.3.1. Diagnostic Interviews

The sections on SAD and generalized anxiety disorder in the Anxiety Disorders Interview Schedule (ADIS-IV; [42]) and the Mini International Neuropsychiatric Interview Version 5.0.0 (MINI-5; [44,45]) were administered in person. The objective of these semi-structured interviews was to establish a reliable diagnosis of SAD and facilitate differential diagnosis with generalized anxiety disorder. The ADIS-IV provides a clinical severity score ranging from 0 (no symptom, distress, or interference) to 8 (severe symptoms, distress, or interference). A score of 4 and above warrants a diagnosis. Following the publication of the DSM-5, the diagnostic criteria for SAD were slightly revised, but diagnoses established with the ADIS-IV remain valid for adults according to DSM-5 criteria.

#### 2.3.2. Self-Report Measures of Anxiety and Social Desirability

The Liebowitz Social Anxiety Scale (LSAS; [46,47]) was administered to document levels of social anxiety in participants. The Balanced Inventory of Desirable Responding Short Form (BIDR-16; [48,49]) was used to measure social desirability and control for its impact, if necessary. These questionnaires were administered prior to the immersion in the control VE.

#### 2.3.3. VR-Related Measures

To control for possible confounds and document participants’ experiences, immersive tendencies, unwanted negative side effects induced by immersions in VR, and the sense of presence were measured. Participants completed the Immersive Tendencies Questionnaire (ITQ; [50,51]) at pretest to assess predisposition to react to mediated environments with focus, involvement and emotions [18,20,22,24]. The Simulator Sickness Questionnaire (SSQ; [51,52]) was administered at pretest, after the immersion in the control VE and after the immersion in the experimental VE to document potential unwanted negative side effects induced by immersions in VR, such as nausea or eye strain [20,52]. The SSQ was scored following Bouchard et al.’s [52] recommendations, and SSQ-Total raw scores are reported. The sense of presence was measured by the Independent Television Commission—Sense of Presence Inventory (ITC-SOPI; [51,53]), administered after each immersion, to document spatial presence (feeling of really being in the VEs as opposed to being in a research lab), engagement (feeling involved and enjoying the immersion), ecological validity (feeling that the VE was lifelike and real), and negative effects (feeling adverse physiological reactions).

#### 2.3.4. Measure of Anxiety (Main Dependent Variable)

The State–Trait Inventory for Cognitive and Somatic Anxiety (STICSA; [43,54]) was used to assess cognitive and somatic symptoms of anxiety experienced when completing the instrument (state anxiety) and in general (trait anxiety). The state anxiety subscale of the STICSA was administered after the immersions in the control VE and the experimental VE. A Subjective Unit of Disturbance Scale for Anxiety (SUDS-A; [43,51]) was used to assess the level of state anxiety experienced during the immersions with a Likert scale from 0 to 10 (with anchor points at 0 = Not at all, 2 = Slightly, 5 = Moderately, 8 = A lot, and 10 = Totally). The SUDS-A was administered six times: after completing the task in the control VE, after each of the four experimental tasks in the experimental VE, and at the end of the experimental immersion to assess the overall experience in the corner store.

### 2.4. Materials

The Z800 VR headset from eMagin™ (New York, NY, USA) and a Cube3 from InterSense™ (Billerica, MA, USA) were used for the immersion. The VEs were developed by the UQO’s research team. The control VE consisted of an apartment without stimuli evoking social interactions [53] and was used for two minutes to familiarize participants with using VR and exploring a VE. The experimental VE depicted a convenience store where the participant could explore the different aisles of food, beverages, and other products frequently sold in this type of store (Figure 2). The virtual human clerk of the store was located behind the cash register. For standardization, the clerk’s questions and answers were pre-recorded in French and Spanish.

## 3. Results

### 3.1. Preliminary Statistical Analyses

Prior to the main analyses of the study, preliminary analyses were conducted to confirm main variables respected assumptions for parametric analyses, test whether there were significant pre-immersion differences between Conditions and Countries on selection, descriptive, control, dependent, and exploratory variables, and test whether control variables contributed to the main findings. Data exploration was conducted to ensure that assumptions of parametric analyses for the two dependent variables (i.e., repeated-measures ANOVAs with STICSA and SUDS-A) were met. The Shapiro–Wilks omnibus normality test revealed slight normality irregularities in the distribution of some variables. The assumption of homogeneity was verified using Levene’s tests. The results were significant (*F* = 5.55, *p* < 0.01) and revealed that variances were not homogeneous. Two univariate extreme outliers were detected using Z-scores greater than ± −3.29 (*p* < 0.001, bilateral). No extreme multivariate outlier was detected using the Mahalanobis distance method (χ^2^(7) = 24.32, *p* < 0.001). The results remained the same when univariate extreme outliers were removed from the analyses. For statistical power purposes, these outliers were retained in this study and non-parametric tests were also performed. Non-parametric tests fully corroborated results of the parametric tests, and therefore, parametric tests are reported for simplicity.

Univariate analyses of variance were performed with ADIS-IV and LSAS scores (see Table 1) to explore potential differences before conducting the main analyses. For the ADIS-IV, results showed a significant difference for the Condition effect, with participants with SAD reporting moderately severe to severe social anxiety and participants without SAD (controls) showing ADIS-IV scores largely below the clinical threshold for social anxiety. No significant differences were found for the Country effect and the Condition X Country interaction on ADIS-IV. Participants with SAD reported experiencing more anxiety, according to LSAS than control participants. According to LSAS scores, participants with SAD reported on average marked social anxiety (the clinical threshold score is 55), and control participants had a level of social anxiety considered below the clinical threshold. No significant differences were found for the Country effect and the Condition X Country interaction on LSAS scores. Chi-square analyses and analyses of variance were performed with the descriptive variables age, sex, number of children, religion, previous experience with VR, and prior experience with psychotherapy. As reported in Table 1, control participants were significantly more familiar with VR than participants with SAD, and participants with SAD reported having received more psychotherapy in the past. When comparing the two countries, the results revealed that participants from Spain had more children, that the Catholic religion was more prevalent among Canadians (Atheism was more prevalent among Spaniards), and that Canadians reported having received more psychotherapy in the past than Spaniards.

The control variables included measures of social desirability (BIDR-16), immersive tendencies (ITQ) and unwanted negative side effects (SSQ). One-way ANOVAs were carried out for pre-experiment scores on BIDR-16 and ITQ (see Table 1) and repeated-measures ANOVAs were performed for SSQ scores (see Table 2) to rule out unexpected differences between Conditions and Countries. Results showed no significant difference in social desirability scores or immersive tendencies between Conditions and Countries. The potential impact of the control variables on the main analyses was also examined by introducing each of them as a covariable in repeated-measures ANVOVAs. Statistical results for the Time X Condition interactions for the main statistical analyses remained significant after adding BIDR-16 or ITQ as a covariable in ANCOVAs; their contribution was not statistically significant when conducting regressions to predict changes in anxiety (results not shown). For SSQ-Total raw scores, the Condition effect (*F*(2, 126) = 16.04, *p* < 0.001, *η*^2^ = 0.20) and the Time X Country (*F*(2, 126) = 4.33, *p* < 0.05, *η*^2^ = 0.06) interaction were significant. Participants with SAD appeared to experience significantly more unwanted negative side effects than non-socially anxious participants for all three measurement times (i.e., before and after the immersion in the control environment and after immersion in the experimental environment) (see Table 2). Furthermore, the Time X Country interaction was significant and revealed that intensity of unwanted negative side effects reported by Canadians increased over time, whereas the scores of Spaniards decreased over time. The Main Time effect (*F*(2, 126) = 0.26, *p* = n.s., *η*^2^ = 0.00), the Country effect (*F*(2, 126) = 0.07, *p* = n.s., *η*^2^ = 0.00), Time X Condition interaction (*F*(2, 126) = 0.13, *p* = n.s., *η*^2^ = 0.00), and Time X Condition X Country interaction (*F*(2, 126) = 0.34, *p* = n.s., *η*^2^ = 0.01) were not statistically significant. The impact of SSQ scores on the main dependent variables was further examined with regressions, and SSQ-Total raw scores did not significantly predict changes in anxiety.

Measures of presence (ITC-SOPI) were also collected on an exploratory basis and subjected to 2 Times (Control and Experimental immersions) X Conditions X 2 Countries repeated-measures ANOVAs. Results show higher presence in the experimental environment for the Spatial Presence subscale (*F*(1, 63) = 229.67, *p* < 0.001, *η*^2^ = 0.79), the Engagement subscale (*F*(1, 64) = 37.71, *p* < 0.001, *η*^2^ = 0.37), and the Ecological validity subscale (*F*(1, 64) = 8.1, *p* < 0.01, *η*^2^ = 0.11). No Condition main effect was found for Spatial Presence (*F*(1, 63) = 0.34, *p* = n.s., *η*^2^ = 0.0), Engagement (*F*(1, 64) = 2.46, *p* = n.s., *η*^2^ = 0.04), or Ecological validity (*F*(1, 64) = 1.1, *p* = n.s., *η*^2^ = 0.02). Differences reached statistical significance for the Country effect on Spatial Presence (*F*(1, 63) = 4.62, *p* < 0.05, *η*^2^ = 0.07) and Ecological validity (*F*(1, 64) = 10.39, *p* < 0.01, *η*^2^ = 0.14), but not Engagement (*F*(1, 64) = 0.89, *p* = n.s., *η*^2^ = 0.01). Among all interactions, only Time X Country for Spatial Presence reached statistical significance (*F*(1, 63) = 4.19, *p* < 0.05, *η*^2^ = 0.07). Presence was rated as higher in the experimental VE compared to the control one and in Canadians compared to Spaniards. Presence was not a statistically significant predictor of change in anxiety from the control to the experimental VEs. 

### 3.2. Main Statistical Analyses

For STICSA, a 2 Times (Control and Experimental immersions) X 2 Conditions (participants with and without SAD) X 2 Countries (Canada and Spain) repeated-measures ANOVA was performed. A similar analysis was conducted for SUDS-A using ratings from the immersion in the control environment and the overall assessment of participants’ anxiety during the corner store immersion. A Bonferroni correction was applied to control the risk of Type I errors. The probability level to declare an effect statistically significant was set at 0.025 (i.e., 0.05/2 variables). To refine the analyses and examine each awkward social interaction task, a 5 Times (Control environment, Task, Task 1, Task 2, Task 3, and Task 4) X 2 Conditions (participants with and without SAD) X 2 Countries (Canada and Spain) repeated-measures ANOVA was performed with SUDS-A measures, followed by interaction contrast analyses comparing each awkward social interaction with the control environment. Greenhouse–Geisser correction to the degrees of freedom was applied to control for sphericity. STICSA and SUDS-A scores were significantly correlated for assessing anxiety in the control environment (*r* = 0.45, *p* < 0.001) and in the experimental condition (*r* = 0.77, *p* < 0.001).

For STICSA (Table 3), the results showed a significant difference between the Time effect and the Condition effect. No significant differences were found for the Country effect. The Time X Condition interaction was found to be significant, whereas the Time X Country and Time X Condition X Country interactions were not significant. Essentially, STICSA scores increased significantly more for participants with SAD in the immersion in the corner store compared to control participants.

Results for SUDS-A (Table 3) comparing anxiety from the immersion in the control environment and overall immersion in the experimental environment in the corner store indicated a significant difference for the Time effect, for the Condition effect and, most importantly, for the Time X Condition interaction. No significant difference in anxiety was identified for the Country factor, the Time X Country interaction, or the Time X Condition X Country interaction. Results with this measure were fully consistent with those from STICSA.

Experienced anxiety for each awkward social interaction task is reported in Table 4 and Figure 3. The repeated-measures ANOVA revealed a significant effect for the Time factor (*F*(3.32, 212.71) = 28.99, *p* < 0.001; *η*^2^ = 0.31), the Condition factor (*F*(1, 64) = 49.17, *p* < 0.001; *η*^2^ = 0.43), and the Time X Condition interaction (*F*(3.32, 212.71) = 11.70, *p* < 0.001; *η*^2^ =0.16). No significant differences were found for the Country factor (*F*(1, 64) = 2.23, *p* = 0.14; *η*^2^ = 0.03), the Time X Country interaction (*F*(3.32, 212.71) =0.06, *p* = 0.99; *η*^2^ = 0.00), or the Time X Condition X Country interaction (*F*(3.32, 212.71) = 1.78, *p* = 0.15; *η*^2^ = 0.03). These analyses confirmed that four tasks in the experimental VE induced significantly higher anxiety than exploring the control VE for participants with SAD compared to control participants. To decompose the Condition X Time interaction, orthogonal interaction contrasts were performed comparing Conditions’ anxiety ratings at the end of the immersion in the control VE (apartment) and after each task performed by participants in the experimental VE (see Table 4). After looking at the results (see Figure 2), an additional set of contrasts was performed a posteriori to compare SUDS-A of only participants with SAD after the second task and after the other three tasks. The only statistically significant difference was lower anxiety scores at the second task compared to the first task (Time *F*(1, 29) = 8.09, *p* < 0.01; *η*^2^ = 0.22). The Time X Country interaction was close to statistical significance (Interaction *F*(1, 29) = 3.72, *p* = 0.06; *η*^2^ = 0.11). The comparison between the second and the third task was close to statistical significance (Time *F*(1, 29) = 4.12, *p* = 0.05; *η*^2^ = 0.13).

## 4. Discussion

The objective of this study was to compare anxious reactions of people with and without generalized SAD during immersion in VR. Participants were prompted to engage in silly and awkward social interactions with a virtual character. Due to a paucity of information on potential cultural differences in use of VR for clinical applications [30], a multisite study was conducted jointly by clinical research centers in Canada and Spain. A virtual environment relatively relatable to Western/European cultures was created to depict a corner store where the participants were prompted to engage socially with the clerk. It was hypothesized that engaging in awkward social interactions would induce a higher level of anxiety in people with generalized SAD than in people without SAD. No hypothesis was formulated for comparisons across countries.

Analyses confirmed the hypothesis and effect sizes were large; all tasks performed in the experimental VE generated more anxiety than the task of exploring the control VE for participants with SAD compared to control participants. Exploratory analyses of participants from Canada and Spain did not reveal significant differences in anxiety response to the VEs. Statistically significant differences were observed when comparing the immersion in the corner store as a whole with the STICSA and the SUDS-A. Immersion in the experimental VE resulted in a statistically significant increase of more than six points for STICSA and more than three points on the Likert scale of the SUDS-A. There is no typical anxiety level that can be used to gauge whether stimuli are appropriate for optimal therapeutic exposure (see [17,18,22]), because anxiety level will vary within and between sessions depending on tasks agreed upon with patients. However, stimuli that could not elicit anxiety because they are virtual would not be useful for exposure.

The hypothesis was also confirmed when comparing specific tasks performed during the two immersions. Participants with SAD showed an increase in anxiety levels, while the scores for the non-clinical sample remained relatively unchanged or even lower than in the control VE. The second task, an impromptu request to name as many Michael Jackson songs as possible, was the least anxiogenic of the tasks performed in the VE. This may be because performing poorly on this task does not elicit the same fear of ridicule to participants with SAD compared to other tasks. Naming songs from a popular artist may be more closely connected to general knowledge compared to the other tasks, which related more to the participants themselves on a personal level. It is important to underline that, in the case of SAD compared to specific phobias, the environment itself may not be sufficient to elicit anxiety about being judged by others. The environment provides a context to engage in exposure to feared social interactions. In clinical practice, clinicians must be creative and design exposure tasks adapted to each patient (e.g., asking the clerk for a date). Further, VEs’ ability to facilitate exposure for SAD may involve variables such as realness and quality of the social interactions with the virtual character (e.g., body language, facial expression). Such features are represented most accurately in 360° video recordings of actual social interactions. Research using 360° videos for VR immersion for SAD are currently ongoing (e.g., [55,56]). Della Libera et al. [56] have shown that immersions in VEs created with 360° videos can be used to conduct exposure exercises for people with SAD and randomized control trials are currently being conducted (e.g., [55]). Using 360° images and video recordings would ensure using highly realistic virtual stimuli and characters, which was not the case with the real-time 3D-rendered synthetic stimuli used in the current study. However, currently, it is hardly possible to create fluid and dynamic complex unexpected interactions using pre-recorded videos.

Participants with SAD reported significantly higher scores on the SSQ before the experiment, after immersion in the control environment, and after immersion in the experimental environment. These results are consistent with Bouchard et al.’s [52] contention that SSQ scores can be inflated by anxiety symptoms. Nevertheless, in general all participants tolerated the VR experience very well. No participant felt uncomfortable enough to have to stop the experiment. All participants reported feeling more present and more psychologically involved, and considered the situation to be more real in the immersion in the experimental environment than in the control environment. The feeling of presence is an important aspect to consider during clinical immersions in VR, as it allows virtual stimuli to generate emotional reactions that are expected or required in psychotherapy [57]. A minimum level of presence may be required to trigger a feeling of anxiety in users immersed in anxiety-provoking stimuli, although presence is not sufficient to ensure the effectiveness of exposure-based CBT treatments [22].

Interestingly, the analyses highlighted differences in presence and immersive tendencies between Canadians and Spaniards. Scores on the ITQ measuring one’s self-reported ability to ignore the distracting effects of the surrounding environment (Focus subscale) were higher in Canadian participants and lower in terms of the commitment to computer and video games (Game subscale). Scores on the ITQ are expected to predict the strength of the feeling of presence experienced in VR. The illusion of actually being in the apartment and the corner store (spatial presence) and the perceived realism of the VEs (ecological validity) were stronger in the Canadian sample than in the Spanish sample. The differences in presence may be associated with differences in immersive tendencies or differences in cultural congruence of the VEs as the VEs were designed in Canada and are implicitly representative of Canadian culture. The apartment decor and the corner store layout may be less typical of Spanish culture. Although products in the corner store were generic and no real brands were replicated, how they were placed on shelves and displayed were typical of Canadian culture. A study in Mexico, by González-Perellon et al. [58], used a VE developed in Canada where cultural incongruences were more important than the current study and did report problems in presence. Van der Sluis et al. [59] assessed the level of familiarity participants from Dublin and Tokyo attributed to a shopping center VE that was atypical of both cultures. Despite cultural incongruences, both participants from Dublin and Tokyo reported a high level of familiarity with the VE. Jan et al. [60] examined how native speakers of American English, Mexican Spanish, and Arabic perceive social interactions among virtual characters and found differences in perceived realism of the interactions, which may suggest the virtual character could have contributed to the observed differences in presence. Cultural differences between Canada and Spain are not as pronounced as between either of these two countries and East Asia, for example [61], so it is possible to assume larger effects when comparing other cultures. Nevertheless, the experimental immersion induced the expected reaction on the main dependent variables in participants from both countries. More research is needed to document cultural differences in presence and emotions elicited by VEs designed for the assessment and treatment of mental disorders [30], paying attention to large and to more subtle differences among countries and to cultural differences within countries (e.g., with Aboriginals [62]).

As for the limitations of this study, the fact we sought to recruit only native French-speaking and Spanish-speaking participants made recruitment difficult. Secondly, three clinicians from different cultural backgrounds selected and assessed the eligibility of potential participants, which may have biased the scores reported in the ADIS-IV. Indeed, Tseng et al. [63] support this hypothesis by suggesting that the diagnostic opinion of mental health professionals is influenced by culture. The study would have gained in specificity if a group of participants with performance-only SAD had been included. An attempt was made to include participants from both types of SAD [51]; however, the number was too small in the Spanish sample and no participants were recruited in the Canadian sample. An immersion in the virtual corner store without engaging in any awkward social interaction would have also been methodologically revealing. It is important to highlight that statistical differences were found between participants from Canada and Spain (number of children, religion, having received psychotherapy) and between people with and without SAD (previous experience with VR, having received psychotherapy). Finally, studies addressing cultural differences should explore additional control variables, such as contrasting cultures that differ in architecture, urbanism, population density, and social dynamics.

## 5. Conclusions

This study contributes to the advancement of knowledge by illustrating how engaging in behaviors that are typically feared by people with generalized SAD can elicit anxiety and be used in exposure-based CBT. It highlights that, in the case of SAD compared to specific phobias, it is not only the virtual stimuli that matter. Behavioral tasks when immersed in VR can be used to control exposure [22], which allows psychotherapists to be creative and active during the immersions to engage patients in numerous awkward and silly situations. The study also suggests that slight cultural differences may not be very detrimental when using VEs developed to treat mental disorders. It is possible to include the cultural incongruences into the narrative that introduces immersion to facilitate presence. For example, clinicians could introduce the narrative by saying that the patient from Asia is on vacation in Canada (or another country) and has to visit this small store to purchase food. Differences in brand of food, display of items, background music, and attitude of the staff would then have less impact on perceived realism for that patient.

## Figures and Tables

**Figure 1 jcm-12-04525-f001:**
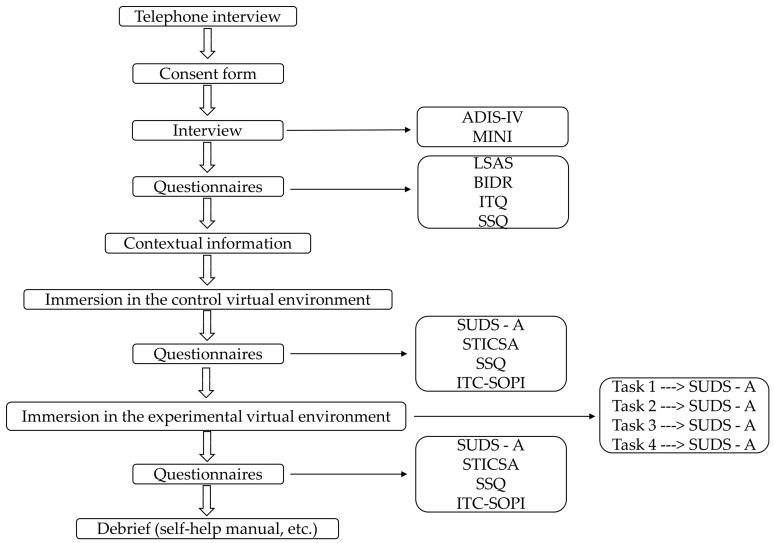
Flowchart illustration of study design as it was conducted in both sites.

**Figure 2 jcm-12-04525-f002:**
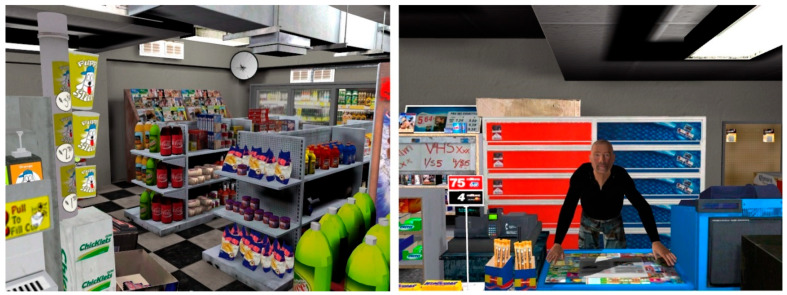
Screenshot of the experimental virtual environment (convenience store) and the clerk.

**Figure 3 jcm-12-04525-f003:**
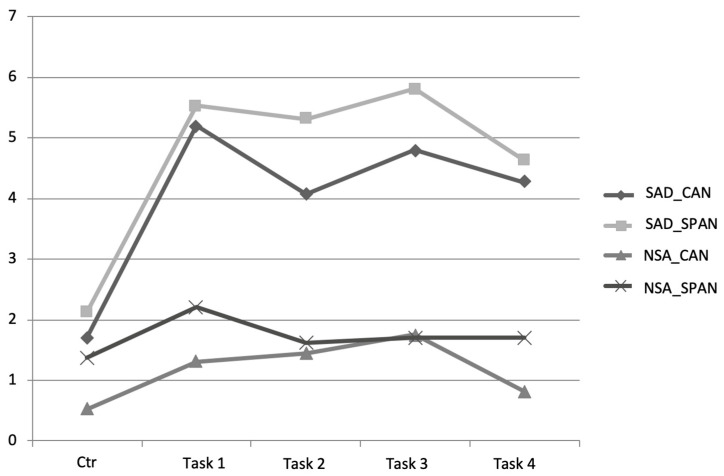
Mean anxiety scores reported on the SUDS-A during the immersion in the control environment and the four tasks performed in the experimental environment according to the condition and country of the participants. Note: SUDS-A = Subjective Units of Disturbance Scale for Anxiety; SAD = Social anxiety disorder; NSA = Non-socially anxious; CAN = Canadian; SPAN = Spanish; Ctr = Anxiety reported during the immersion in the control virtual environment; Task 1 = Informal interaction; Task 2 = Formal interaction (performance); Task 3 = Assertiveness; Task 4 = Being observed.

**Table 1 jcm-12-04525-t001:** Results of sociodemographic characteristics, ADIS-IV severity ratings of SAD, anxiety, social desirability, and immersive tendencies.

	Canada		Spain		Condition Effect	Country Effect	Interaction Condition X Country
	PSAD (*n* = 15)	Control (*n* = 17)	PSAD (*n* = 16)	Control (*n* = 21)			
Age †	35.33 (16.84)	24.88 (5.12)	23.06 (4.12)	29.81 (10.35)	0.56 (0.01)	2.19 (0.03)	12.02 ** (0.16)
Sex ^+^					0.48	0.95	N.A.
Male	46.70%	23.50%	43.80%	47.40%			
Female	53.30%	76.50%	56.30%	52.40%			
N children †	0.33 (0.49)	0.06 (0.24)	1.00 (0.00)	1.24 (0.44)	0.05 (0.00)	116.29 *** (0.64)	8.97 ** (0.12)
Religion ^++^					1.26	25.99 ***	N.A.
Catholic	73.30%	70.60%	12.50%	15%			
Atheist	26.70%	29.40%	87.50%	85%			
Prev. exp. VR ^+^					7.07 **	0.15	N.A.
No	80%	58.80%	93.80%	57.10%			
Yes	20%	41.20%	6.30%	42.90%			
Prev. Psychot.					9.07 **	5.32 *	N.A.
No	46.70%	76.50%	68.80%	100%			
Yes	53.30%	23.50%	31.30%	0%			
ADIS-IV †	5.80 (0.82)	1.15 (0.90)	5.56 (0.75)	0.90 (0.96)	487.78 ***	1.30	0.00
LSAS †	76.87 (23.34)	16.09 (11.92)	70.19 (17.66)	19.05 (13.40)	190.18 *** (0.75)	0.21 (0.00)	1.41 (0.02)
BIDR-6 †	80 (20.26)	83.82 (15.46)	84.84 (7.87)	79.43 (8.52)	0.06 (0.00)	0.00 (0.00)	1.96 (0.03)
ITQ †	62.40 (23.67)	70.94 (14.42)	65.38 (15.13)	71.00 (12.54)	3.12 (0.05)	0.14 (0.71)	0.13 (0.00)

Note: Standard deviations are presented in brackets to the right of the means (no s.d. is reported for percentages). The partial *η*^2^ are presented in brackets to the right of the F-values. PSAD = Participants with social anxiety disorder; N = Number; Prev. = Previous; exp. = experience; Psychot. = Psychotherapy; VR = Virtual reality; ADIS-IV = Anxiety Disorders Interview Schedule for DSM-IV; LSAS = Liebowitz Social Anxiety Scale; BIDR-6 = Short transcultural version 6 of the Balanced Inventory of Desirable Responding; ITQ = Immersive Tendencies Questionnaire; N.A. = Not applicable. † = The degrees of freedom for these analyses are 1 and 65; ^+^ = The degrees of freedom for these analyses are 1; ^++^ = The degrees of freedom for these analyses are 2. * = *p* < 0.05; ** = *p* < 0.01; *** = *p* < 0.001.

**Table 2 jcm-12-04525-t002:** Results of unwanted negative side effects induced by immersions in VR (cybersickness) and presence according to time, condition, and participants’ country.

	Canada		Spain	
	PSAD (*n* = 15)	Control (*n* = 17)	PSAD (*n* = 16)	Control (*n* = 21)
Cybersickness				
Pretest	5.67 (6.56)	2.06 (2.33)	7.81 (5.43)	3.14 (2.08)
Ctr. imm.	6.67 (5.38)	2.63 (4.53)	7.00 (6.64)	3.52 (3.16)
Exp. imm.	7.93 (6.68)	3.56 (6.05)	6.50 (4.86)	2.14 (2.22)
Presence:				
Spatial ence				
Pretest				
Ctr. imm.	49.29 (11.82)	50.06 (8.55)	46.94 (11.23)	45.52 (11.47)
Exp. imm.	66.93 (16.93)	71.50 (10.57)	56.69 (9.43)	61.57 (11.94)
Engagement				
Pretest				
Ctr. imm.	40.27 (10.94)	43.88 (6.79)	39.06 (7.24)	40.57 (6.59)
Exp. imm.	44.87 (6.30)	48.75 (7.09)	45.00 (8.67)	46.67 (7.76)
Ecological validity				
Pretest				
Ctr. imm.	18.13 (7.06)	17.53 (3.14)	13.44 (3.27)	15.24 (3.86)
Exp. imm.	18.07 (4.80)	19.38 (3.58)	16.56 (2.73)	17.48 (3.82)
Negative effects				
Pretest				
Ctr. imm.	10.00 (4.05)	10.24 (6.35)	11.06 (4.78)	11.00 (4.52)
Exp. Imm	10.33 (3.48)	10.13 (4.87)	12.13 (5.28)	9.57 (3.53)

**Table 3 jcm-12-04525-t003:** Results of repeated-measures ANOVAs on the dependent variables.

	Canada		Spain		Time Effect	Condition Effect	Country Effect	Time X Condition	Time X Country	Time X Condition X Country
	PSAD (*n* = 15)	Control (*n* = 17)	PSAD (*n* = 16)	Control (*n* = 21)						
STICSA					10.65 ** (0.15)	21.27 *** (0.25)	0.61 (0.01)	14.30 *** (0.19)	0.55 (0.01)	0.05 (0.00)
Ctr. imm.	30.21 (8.27)	23.44 (5.37)	30.56 (10.90)	25.00 (6.69)						
Exp. imm.	35.71 (13.29)	22.50 (1.71)	37.81 (14.44)	25.00 (7.60)						
SUDS-A					74.92 *** (0.54)	32.74 *** (0.34)	2.85 (0.04)	36.25 *** (0.37)	0.03 (0.00)	0.14 (0.00)
Ctr. imm.	1.70 (2.07)	0.57 (0.82)	2.13 (2.19)	1.38 (1.66)						
Exp. imm.	4.71 (2.47)	1.17 (0.98)	5.36 (2.21)	1.90 (1.36)						

Note: Standard deviations are presented in brackets to the right of the means. The partial *η*^2^ are presented in brackets to the right of the F-values. PSAD = Participants with social anxiety disorder; Ctr. imm. = Immersion in the control VE; Exp. imm. = Immersion in the experimental VE; STICSA = State–Trait Inventory for Cognitive and Somatic Anxiety; SUDS-A = Subjective Unit of Disturbance Scale for Anxiety. ** = *p* < 0.01; *** = *p* < 0.001.

**Table 4 jcm-12-04525-t004:** Anxiety ratings (SUDS-A) after immersion in the control virtual environment and performing four awkward social interactions tasks in a virtual corner store (experimental environment) for adults with and without social anxiety from Canada and Spain.

	Canada		Spain		
	PSAD (*n* = 15)	Control (*n* = 17)	PSAD (*n* = 16)	Control (*n* = 21)	Interaction Contrast with Ctrl. Immersion (Partial *η*)
Ctrl. Immersion	1.70 (2.07)	0.57 (0.82)	2.13 (2.19)	1.38 (1.66)	N.A.
Task 1	5.20 (2.67)	1.31 (2.21)	5.53 (1.96)	2.21 (1.81)	27.16 *** (0.30)
Task 2	4.07 (2.60)	1.44 (1.62)	5.31 (2.33)	1.62 (1.47)	20.51 *** (0.24)
Task 3	4.80 (2.76)	1.75 (1.69)	5.81 (2.69)	1.71 (1.55)	25.66 *** (0.29)
Task 4	4.27 (2.55)	0.81 (1.05)	4.63 (3.44)	1.00 (1.62)	23.04 *** (0.27)

Note: Standard deviations are presented in brackets to the right of the means. The partial *η*^2^ are presented in brackets to the right of the F-values for the interaction contrasts. PSAD = Participants with social anxiety disorder; Control = Participants without social anxiety disorder; Ctr. immerssion = Immersion in the control virtual environment; N.A. = Not applicable; Task 1 = Informal interaction; Task 2 = Formal interaction (performance); Task 3 = Assertiveness; Task 4 = Being observed. *** = *p* < 0.001.

## Data Availability

The dataset is not publicly available due to privacy and ethical restrictions. The data for this study are available upon request addressed directly to the Research Ethics Boards of the lead institution (comite.ethique@uqo.ca), which must first approve the request. If the request is approved, anonymized data supporting the conclusions of this manuscript will be made available by the corresponding author.

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
