# Peer review of "Engaging in Awkward Social Interactions in a Virtual Environment Designed for Exposure-Based Psychotherapy for People with Generalized Social Anxiety Disorder: An International Multisite Study"

_jcm, 2023, doi:10.3390/jcm12134525_

Round 1
Reviewer 1 Report
In this paper, authors test if engaging in silly social interactions in a VE could induce anxiety in individuals with generalized SAD and also exploit potential cultural differences in anxiety intensity by employing comparative experiments which include population from different countries, Canada and Spain. It’s a pretty well-organized manuscript with novel ideas including studying the effects of virtual environment on people who are suffering from social anxiety disorder and differences across countries in clinical VR applications. However, I have some suggestions which might help in improving this paper as follows:
1. The motivations for cross-countries comparisons can be stated more in detail so that authors can demonstrate the significance of showing the cultural differences.
2. Significant result can be highlighted in bold in all of tables. If Table 1 is adjusted to be in one page, that would be better.
3. In Table 1, for comparison test (different countries, genders), adding columns (or rows) for the difference of results will be more easy to read.
4. For repeated Anova measures in the part of ‘Main statistical analyses’, some assumptions may be needed to ensure the correctness.
5. In some of the tables, standard deviations are shown in brackets to the right of the means. More explanations of sd would be helpful or it would just be displaying data.
6. This study seems to have more to explore if some controlling variables are changed. So conclusion could contain more about the expectations of future study.
Reviewer 2 Report
My comments are attached.

Reviewer 3 Report
The study involved a virtual reality simulation of a series of awkward simulations which was completed by a sample of individuals with social anxiety disorder (SAD) and healthy controls in Canada and Spain. SAD individuals exhibited greater increases in anxiety during the virtual reality exposure compared to healthy controls in both the Canadian and Spanish samples. The results are interesting and have potential implications for virtual reality exposure treatment for SAD. I had some concerns outlined below.
Major concerns:
I wonder if “awkward” is a better description than “silly” for the virtual situations.
While it was shown that participants with social anxiety exhibited greater increases in anxiety compared to healthy controls, can we conclude that the level of anxiety experienced by participants was appropriate for exposure treatments to optimize therapeutic benefits? What would typical SUDS-A scores be for in vivo exposures during clinical treatment?
Page 3: “being assertive, making blunders or being involved in silly situations [41].” This needs to be elaborated as the cited reference doesn’t discuss silly situations (it discusses informal situations but these don’t qualify as silly, e.g. going to a party or meeting strangers).
Given the number of ANOVA and ANCOVA models presented in the preliminary analyses section, it would be helpful to describe more clearly how these specific models were chosen.
Minor concerns:
Page 2: “Social Anxiety Disorder (SAD) is the third most common mental disorder in Western society after depression and alcohol dependence [1]”; The cited reference doesn’t seem to state this. Also, prevalence of specific phobia appears to be higher than SAD in the National Comorbidity Study – Revised.
Abstract: “The impact of each socially threatening task performed during the experimental immersion was statistically significant.” Should this specify “impact on anxiety”?
Abstract: “However, spatial presence and ecological validity were higher in Canadians than in Spaniards.” Spatial presence and ecological validity may warrant brief explanatory phrases in the abstract (and fuller explanation in the introduction) as they may not be familiar to all readers in the context of VR.
Page 4: “The purpose of this instruction was to provide a narrative that would help create a context to facilitate presence and make one some of the tasks more anxiogenic”; There is a typo in this sentence (“one some”).
Page 5: “diagnostic established with the ADIS-IV remain valid for adults according to DSM-5 criteria”; Typo (should be “diagnosis”)
Page 5: I would recommend further explanation/description of the VR-related measures (regarding immersion, sickness, and sense of presence), as these may not be familiar to readers.
The use of the term “Condition” rather than “Group” to refer to SAD vs. controls may be confusing given that “Condition” often refers to an experimental condition.
Page 15: “Cultural differences between Canada and Spain are not as pronounced as between any of these two countries and East Asia, for example”; “any” should be “either”
There were a few minor errors/typos
Round 2
Reviewer 2 Report
authors have addressed all raised comments